# Force-Induced Alignment of Nanofibrillated Bacterial Cellulose for the Enhancement of Cellulose Composite Macrofibers

**DOI:** 10.3390/ijms25010069

**Published:** 2023-12-20

**Authors:** Ruochun Wang, Tetsuo Fujie, Hiroyuki Itaya, Naoki Wada, Kenji Takahashi

**Affiliations:** 1Graduate School of Natural Science and Technology, Kanazawa University, Kanazawa 920-1192, Japan; wangruochun@stu.kanazawa-u.ac.jp; 2Institute of Science and Engineering, Kanazawa University, Kanazawa 920-1192, Japan; tfujie@p.kanazawa-u.ac.jp (T.F.); hitaya@se.kanazawa-u.ac.jp (H.I.); naoki-wada@se.kanazawa-u.ac.jp (N.W.)

**Keywords:** bacterial cellulose, dry jet-wet spinning, wet stretching, alignment, mechanical properties, ionic liquid

## Abstract

Bacterial cellulose, as an important renewable bioresource, exhibits excellent mechanical properties along with intrinsic biodegradability. It is expected to replace non-degradable plastics and reduce severe environmental pollution. In this study, using dry jet-wet spinning and stretching methods, we fabricate cellulose composite macrofibers using nanofibrillated bacterial cellulose (BCNFs) which were obtained by agitated fermentation. Ionic liquid (IL) was used as a solvent to perform wet spinning. In this process, force-induced alignment of BCNFs was applied to enhance the mechanical properties of the macrofibers. The results of scanning electron microscopy revealed the well-aligned structure of BCNF along the fiber axis. The fiber prepared with an extrusion rate of 30 m min^−1^ and a stretching ratio of 46% exhibited a strength of 174 MPa and a Young’s modulus of 13.7 GPa. In addition, we investigated the co-spinning of carboxymethyl cellulose-containing BCNF with chitosan using IL as a “container”, which indicated the compatibility of BCNFs with other polysaccharides. Recycling of the ionic liquid was also verified to validate the sustainability of our strategy. This study provides a scalable method to fabricate bacterial cellulose composite fibers, which can be applied in the textile or biomaterial industries with further functionalization.

## 1. Introduction

Plastic waste has become a global concern because of its substantial accumulation in the environment and the significant risk to various ecosystems and living organisms [1,2]. In the last decade, environmental pollution due to plastic waste has continuously increased. Plastic production is expected to reach 500 million tons in 2025, and approximately 60% of it will enter our environment as waste [3]. Therefore, there is a concerted effort to prioritize and expedite the development of biomaterials to reduce the consumption of petroleum-based polymers [4,5,6,7]. Textiles and fibers, which are globally high in demand, are also required to develop green and sustainable products [8]. 

Cellulose is the most abundant and commonly utilized natural polymer for textile fabrication because of its renewability, great mechanical strength, hydrophilicity, and good dyeability [9,10,11,12,13]. It is a polysaccharide containing long linear β-(1,4)-_D_-glucose units with a large amount of hydroxyl groups [14,15]. This attribute leads to the formation of a huge hydrogen-bonding network among cellulose molecules, providing this natural polymer with high crystallinity and excellent mechanical properties [16]. Further, the utilization of regenerated cellulose, the product of cellulose dissolution and precipitation, and cellulose derivatives also account for a considerable proportion in textile industry because they can be easily processed via spinning to fabricate synthetic fibers [17]. Viscose rayon and lyocell have become the most popular types of regenerated fibers in 2020 due to their softness and wearing comfort [18]. Thus far, cellulose, regenerated cellulose and their derivatives have been widely used in our daily life [19,20]. However, in the light of the growing demand for biomaterials and green textiles, we need to find more accessible sources to afford such extensive consumption of cellulose. In this context, bacterial cellulose (BC) has attracted great attention due to its short production period [21,22,23]. Unlike plant-derived cellulose, BC does not require excess chemical and physical pretreatment to remove other impurities, such as hemicellulose, lignin, and pectin [24]. Although BC has an identical chemical structure to plant-derived cellulose, it also exhibits a natural nanostructure and a high aspect ratio. It is a desirable raw material for the development of advanced materials [25,26,27,28]. BC is mainly produced using two approaches: static and agitated fermentation [29]. In static fermentation, a gelatinous pellicle can be obtained at the interface of air and culture media, while agitated fermentation is used to prepare BC dispersions. Generally, the BC pellicle produced through static fermentation has a higher degree of polymerization and crystallinity, resulting in good mechanical properties. However, agitated fermentation can be easily amplified for large-scale production and the obtained BC exhibits better water holding capacity as compared to static cultivation [30].

Although BC exhibits such advantages, the mechanical properties of the material products are not only affected by the BC itself. The axial tensile strength and the elastic modulus of single BC nanofibers can reach up to 1–3 GPa and 120–220 GPa, respectively [31,32]. However, these excellent performances are lost when BC nanofibers are assembled from the nanoscale to the macroscale. During this transition, the arrangement and stacking of molecules and nanofibers determine the performance of the materials. Precise alignment and densification of BC significantly contribute to reducing the generation of defects and helping in conducting stress, thus providing the material with unique properties and overcoming its weak mechanical properties [33,34,35]. In 2017, Hu et al. fabricated a super strong BC macrofiber using the wet stretching and twisting methods [36]. A BC wet pellicle was stretched to 30% deformation, followed by immediate twisting into fiber and drying under 90 °C. With a well-aligned interior structure, the fibers exhibited a high tensile strength of 826 MPa and a Young’s modulus of 65.7 GPa. A similar result was reported by Chen et al. [37]. They first soaked a BC pellicle in an N-methyl-2-pyrrolidinone solvent and performed wet stretching to align BC nanofibers. After drying, the product was peeled to obtain ultrathin films that could be twisted into fibers under wet conditions. The fiber not only exhibited high strength (886 MPa), but also had a good toughness of 59.2 MJ m^−3^. The same research group also prepared BC fibers by wet spinning a 2, 2, 6, 6-tetramethylpiperidine-1-oxyl-oxidized BC nanofiber suspension [38]. The fiber was coagulated in acetone and subjected to a two-step stretching process. The tensile strength of the final product was 659 MPa. The above studies reported the use and orientation of BC pellicles, in which BC nanofibers are trapped and stacked more compactly at the nanoscale. However, BC produced from agitated culture showed totally different characterization with BC pellicle [39]. After agitated fermentation, BC is dispersed in culture medium and randomly arranged. It is therefore difficult to assemble this BC slurry into an ordered structure. Although BC slurry can be shaped by drying, the mechanical properties of obtained material are very low and the orientation of BC cannot be controlled after drying due to the completed aggregation of nanofibers [40,41]. Therefore, the use of BC slurry to fabricate a robust material has rarely been reported.

Ionic liquids (ILs) are organic salts with melting temperatures lower than 100 °C [42,43]. Generally, ILs are composed of an organic cation and an organic or inorganic anion. Some special ILs exhibit excellent solubility for polysaccharides which are generally difficult to dissolve in normal solvents [44]. Owing to their high chemical stability and low toxicity, ILs have become popular green media for biomass processing and biomaterial fabrication, including pretreatment, dissolution, catalysis, and others [45]. Further, ILs exhibit negligible vapor pressure, simplifying their recycling process and resulting in cost savings.

Recently, Tajima et al. reported a novel agitated fermentation process to prepare a special nanofibrillated bacterial cellulose (BCNF) [46]. In this process, BCNF slurry is obtained directly by adding cellulose derivatives to disperse the newly produced BC. The produced BCNF exhibited nanostructures with long aspect ratio and adjustable surface hydrophilicity. Hydroxypropyl cellulose-containing BCNF (HPBC) can be dispersed in common organic solvents, and this unique BCNF exhibits great potential for applications in the field of biomaterials and composite materials. Our group has studied the cellulose acetate composites improved by BCNFs and investigated the effects of different processes on mechanical properties [47]. But it is still a difficult problem to directly use BCNF to fabricate materials. ILs can break hydrogen-bonding networks between cellulose chains, which allows the reassembly and alignment of cellulose for preparing robust materials. So here, we considered that BCNFs may be readily wet spun into macrofibers using IL as a solvent and the mechanical properties can be enhanced by stretching. 

In this study, we prepared a BC composite macrofiber from BCNF using a combination of dry jet-wet spinning and wet stretching methods (Figure 1). The BCNFs were freeze-dried and dissolved in 1-ethyl-3-methyl-imidazolium acetate (EmimAc) and dimethyl sulfoxide (DMSO) cosolvent to obtain a homogeneous dope. Spinning was then performed under different conditions, followed by stretching to obtain a well-aligned nanostructure of BCNF. We evaluated the spinnability of the BCNFs and the effect of processing parameters on the mechanical properties of BC composite macrofibers. In addition, utilizing the dissolving ability of ILs to a variety of polysaccharides, it is expected that IL can be used as a “container” to mix BCNFs with other polysaccharides, and the co-spinning of BCNFs with other polysaccharides can be easily achieved.

## 2. Results and Discussion

### 2.1. Rheological Properties of the HPBC Spinning Dope

The rheological properties of spinning dope are significant factors of wet spinning. Excessively high viscosity of the spinning dope makes it difficult to extrude the dope from the cylinder. However, if the viscosity is significantly low, the shaping of the fiber is hindered. Thus, the viscosities of the BCNF spinning dope were evaluated at different temperatures and shear rates. As shown in Figure 1, the viscosity of dope is highly dependent on the temperature and shear rate. The viscosity value of 4 wt% spinning dope sharply reduced from 322 to 20.7 Pa·s with the increase in temperature from 25 to 80 °C. This can be attributed to the intensified movement and unwinding of the cellulose molecular chains at high temperatures. In addition, as the rotation speed increased, the spinning dope exhibited typical shear-thinning characteristics required for spinning. Then, we tried wet spinning from 60 °C to 80 °C under an extrusion rate of 15 m min^−1^. Unfortunately, the diameters of obtained fibers were not uniform at the spinning temperature of 60 °C. And the spinning dope was too dilute to perform wet spinning at 80 °C. Finally, wet spinning was performed at 70 °C.

### 2.2. Dry Jet-Wet Spinning of HPBC

The dry jet-wet spinning process of BCNF is illustrated in Figure 1a. First, HPBC was dissolved in an EmimAc/DMSO mixed solvent and prepared as a 4 wt% spinning dope. It is noteworthy that suitable viscosity for the spinnability of high concentration dope required elevated temperature, which may cause the degradation of BC molecules. Therefore, 4 wt% was chosen as the fitting concentration of dope for wet spinning. IL was used as a solvent and allowed the reassembly of BCNF. Then, the dope was extruded from the cylinder at 70 °C and passed through air gap before entering in the coagulation bath. Here, the dry-jet method was applied to obtain better orientation and enhance the mechanical properties of the BC composite macrofibers. After coagulation, the fiber was washed with water to remove all the IL, yielding gelatinous HPBC fibers (Figure 1b). In the spinning and coagulation processes, BC molecular chains reassembled and crystallized to form new cellulose fibril network. Finally, the fiber was stretched using two rolls at different rotation speeds and air-dried under tension. The BC chains were aggregated and aligned during wet stretching. After drying, the oriented gelatinous fibers shrank and formed HPBC macrofibers with uniform and compact structures (Figure 1c). The extrusion rates and stretching ratios of the partial fiber samples are listed in Table 1. In this strategy, three forces contribute to the alignment of BC molecular chains: shear force from the spinneret, gravity in the air gap, and the stretching force during wet stretching. Through the collaborative action of these forces, a well-aligned microstructure was formed in macrofibers, and the fiber performances were enhanced. 

To understand the crystalline structure transformation of HPBC during spinning, we performed an X-ray diffraction (XRD) analysis of the original HPBC freeze-dried powder and HPBC fibers (Figure 2a). We observed three diffraction peaks in the pattern of the original HPBC powder at 14.76°, 16.80°, and 22.90°, corresponding to the ($1\bar{1}0$), (110) and (200) lattice planes of cellulose crystal I, respectively. This is the typical structure of natural cellulose. However, after spinning, the HPBC fibers exhibit different peaks at 12.60° and 20.53°. These two characteristic peaks correspond to the cellulose crystal II structure. In addition, the peak shift from 1426 to 1419 cm^−1^ in the Fourier-transform infrared (FT-IR) spectra of HPBC powder and HPBC fiber also indicates this crystalline structure transformation (Figure 2b) [48]. Cellulose crystal I in HPBC was broken down by EmimAc and reassembled into crystal II during coagulation in methanol. The crystallinity decreased from 74% to 47% after wet spinning. However, the crystallinity increased slightly after stretching (from 47% to 54%), suggesting that the cellulose chains had a preferred alignment under stretching and formed a larger crystalline area. 

The surface and cross-section morphologies of the HPBC fibers (undrawn) were analyzed using scanning electron microscopy (SEM), and the photos are shown in Figure 2c,d. The fibers exhibited flat and smooth surface, and the cross-section of HPBC fibers showed a circular shape. The diameter of the unstretched HPBC fiber was approximately 140 μm. Further, the HPBC fibers became thinner after being subjected to tensile deformation (Appendix A). The diameters of HPBC fibers with different stretching ratio were shown in Appendix A. After stretching, the fiber diameter gradually decreased to about 109 μm. These results also suggest that the spinning process was successful and smooth.

We also measured the nitrogen content in the HPBC fibers to determine whether EmimAc was present. The results of the elemental analysis (Appendix A) revealed that the nitrogen content in HPBC fiber was significantly low and same with HPBC freeze-dried powder, indicating that there was no IL residue in the HPBC fiber. The water washing process was efficient for obtaining a pure fiber product.

### 2.3. The Aligned Interior Microstructure of the HPBC Fiber

The internal microstructure was investigated using high-resolution SEM. As the fiber surface was very smooth, the alignment of the HPBC nanofiber bundles was difficult to observe. Before the SEM analysis, HPBC fibers were cut open to expose the structure of the inner cellulose nanofiber bundles. Figure 3a shows the microstructure of the undrawn HPBC fibers (Table 1, Entry10). The BC nanofiber bundles exhibited limited alignment and disordered arrangement, suggesting that it was not enough to obtain a well-aligned structure using only shear force and gravity. A certain extent of the orientation of the BCNF was lost during the coagulation and drying process. Therefore, a wet stretching procedure was necessary to obtain a highly ordered arrangement of HPBC. The interior morphologies of the drawn HPBC fibers (Table 1, Entry 12) are shown in Figure 3b,c. The alignment of the HPBC nanofiber bundles along the fiber axis was clearly observed. During the wet stretching process, HPBC nanofibers rotated axially and slipped between each other, resulting in an ordered and dense arrangement. This well-aligned structure was preserved by drying under tension, reducing the number of defects and enhancing the mechanical properties.

### 2.4. Mechanical Properties of the HPBC Fiber

The alignment structure is mainly affected by the extrusion rate and stretching ratio. Thus, we evaluated the mechanical properties of the HPBC fibers using these two variables. Table 1 lists the results for the partial samples, and the complete data are listed in Appendix A. Overall, both the tensile strength and the Young’s modulus of the HPBC fibers slightly increased with the extrusion rate. A high extrusion rate contributed to better ordered arrangement obtained from stronger shear force, resulting in higher mechanical properties. Figure 4 shows the effect of stretching ratio on the mechanical properties of HPBC fibers. Figure 4a shows the increase in the tensile strength and Young’s modulus with increasing stretching ratio. The strength of the HPBC fiber increased from 91 MPa for HPBC-0 to 174 MPa for HPBC-46, which is approximately 2 fold that of HPBC-0. Moreover, Young’s modulus also increased from 8.4 to 13.7 GPa, indicating that the fiber has enhanced rigidity and ability to withstand higher external forces within elastic deformation. The results of mechanical properties for the HPBC-0 and HPBC-13 samples under different extrusion rates showed slight differences. This may be because the stretching ratio was low, and the interior structure of the fiber did not change significantly. From Figure 4b, the stress–strain curves indicate that the yield point is also improved with a higher stretching ratio. Further, as the stretching ratio increased, the density of HPBC macrofibers increased from 1.25 to 1.42 g cm^−3^, indicating that the interior structure gradually became compact. Without stretching, the hydrogen bonding interactions between random HPBC molecules are loose and weak. In sharp contrast, the good alignment and dense structure of stretched HPBC fiber led to increasing numbers of hydrogen bonds and decreasing numbers of defects, resulting in the preferred mechanical properties. These results clearly demonstrate the contribution of good alignment of HPBC to the fiber performance.

Moreover, we explored other factors that could impact the mechanical properties by affecting the interaction between the BC molecular chains, including the washing bath, intrinsic cellulose derivatives, spinneret diameter, and humidity. First, we attempted to coagulate the HPBC fibers in water, methanol, ethanol, and acetone. Unfortunately, the spinning dope could not form fibers in water or acetone. And the solvent exchange rate of ethanol is lower than that of methanol because of its larger molecular size. After the fibers entered the ethanol bath, fibers were conglutinated during washing. Therefore, we collected only two samples in the study using the washing bath. The HPBC fibers were coagulated in methanol and washed separately with water and methanol. As shown in Figure 5a, the HPBC fibers washed with water exhibited better mechanical properties than the fibers washed with methanol. Water molecules can form strong interactions with cellulose molecules compared with methanol. Therefore, water acts as a plasticizer, promoting the sliding between HPBC molecular chains during the stretching process, thus facilitating the alignment of HPBC and generating more hydrogen bonds among cellulose chains. Further, when water evaporates from the fibers, the HPBC molecular chains interact more strongly with each other due to the high surface tension of water, forming a dense structure which could also enhance the mechanical properties.

The BCNF type also had a certain effect on the fiber performance. During the agitated fermentation of BC production, well-dispersed BCNF can be obtained by adding hydroxypropyl cellulose (HPC), carboxymethyl cellulose (CMC), and hydroxyethyl cellulose (HEC), which are denoted as HPBC, CMBC, and HEBC, respectively (Details are shown in Experimental section). We used these three different BCNFs to perform wet spinning and examined the effect of these additional intrinsic cellulose derivatives on the mechanical properties of the obtained macrofibers (Figure 5b). CMBC exhibited a slightly higher strength (160 MPa) and Young’s modulus (12 GPa) than the other two types of BCNF. In our study, CMC was added during the fermentation process and entangled with newly produced BC. Thus, the carboxyl groups of CMC provided the CMBC with a negative charge and generated charge repulsion between the CMBC molecules. This charge repulsion suppressed the random winding of CMBC molecular chains and contributed to the orderly arrangement of BCNF during spinning and wet stretching. Consequently, the mechanical properties of CMBC fibers can be readily enhanced by wet stretching.

Figure 5c shows a comparison of the HPBC fibers extruded from spinnerets with different diameters. Thin HPBC fibers exhibited lower strength with and without stretching. This may be due to the rapid solvent exchange. In the coagulation bath, IL molecules in the fiber were replaced by methanol, and BC regenerated and recrystallized into a gelatinous fiber. However, the regeneration process accelerated when the fiber diameter decreased. In this case, the cellulose molecules were fixed quickly before forming a great hydrogen-bonding network in coagulation and the number of defects increased in fiber, resulting in a reduction in strength.

We further studied the effect of humidity on the mechanical properties of HPBC fibers. The stress–strain curves obtained under different humidities are shown in Figure 5d. Water molecules can enter the interior of cellulose chains and compete for hydrogen bonds with cellulose molecules. In a humid environment, the tensile strength and Young’s modulus decrease owing to the decrease in hydrogen bonds by the existing water. Our results also indicate that the strength decreased to 130 MPa with increase in humidity from 40% to 70%. The Young’s modulus was also decreased. However, after immersion in water for 5 min, the re-wet fibers exhibited a sharp decrease in modulus. Only 1.5 GPa was retained. When the fiber was immersed in water, a large number of water molecules entered into the fiber, thoroughly destroying the original hydrogen-bonding networks in the HPBC fibers, resulting in a sharp reduction in Young’s modulus.

### 2.5. Co-Spinning of CMBC and Chitosan

In our strategy, we used IL as a solvent to prepare a homogeneous spinning dope. However, ionic liquids show excellent solubility not only to cellulose, but also to many polysaccharides. In this case, cellulose-based blend fibers can be easily prepared with other polysaccharides by taking advantage of great dissolving ability of IL. IL can be used as a “container” to blend BCNF with polysaccharides. And the cellulose derivatives in BCNF can contribute to the increased interactions and compatibility with other molecules. In this study, we present an example of co-spinning using CMBC and chitosan (CS). Carboxymethyl cellulose in CMBC can act as an adhesive for connecting BCNF and CS through charge attraction between the carboxyl and amido groups. 

CMBC and CS were first dissolved in EmimAc/DMSO cosolvent and subjected to wet spinning with same procedure of HPBC spinning. The mass ratio was set to 9:1 (CMBC:CS) and the molar ratio of the carboxyl and amido groups was approximately 1:1. However, the viscosity of the CMBC-CS solution was substantially high, which may be due to the charge attraction between carboxyl and amido groups. Thus, dissolving and spinning temperatures were increased to 95 °C. The chemical composition of this blend fiber (CMCS) was investigated by FT-IR (Figure 6a). The strong peak at 1026–1060 cm^−1^ can be assigned to C-O-C linkage of polysaccharides backbone. The wide signal around 3200–3600 cm^−1^ is corresponding to O–H stretching vibration. The characteristic peak at 1596 cm^−1^ in the CMBC spectrum was ascribed to the carboxyl groups. The signal of amide groups was observed at 1649 cm^−1^ (amide I band) and 1561 cm^−1^ (amide II band) in the CS spectrum. Both peaks appear in the spectrum of CMCS fiber, suggesting the existence of two components. The mechanical properties of the blend fibers are shown in Figure 6b. The CMCS fibers also exhibited enhanced tensile strength and Young’s modulus with increasing stretching ratio, which is consistent with the results of HPBC fibers. However, the mechanical strength was not enhanced when the stretching ratio exceeded 26%. CMBC forms strong interactions with CS via charge attraction and hydrogen bonds. Such strong interactions may limit the rotation and slip between polysaccharide chains that occur during wet stretching, and reduce the efficiency of alignment.

### 2.6. Recycling of the IL

ILs are green, non-toxic, but expensive. Therefore, the recycling of ILs is of great significance for controlling production costs. Fortunately, most ILs have low vapor pressures and can be readily purified from common organic solvents. After wet spinning and washing, EmimAc was recycled via rotary evaporation and vacuum distillation. Approximately 95% of the EmimAc was recovered. Recycled EmimAc was characterized by ^1^H nuclear magnetic resonance (NMR) and compared with fresh EmimAc (Figure 7a). From the ^1^H NMR spectra, we can see that the recycled product does not contain any other cellulose-based polymers. A small peak appeared at 2.6 ppm, which was attributed to residual DMSO. It is difficult to remove all DMSO because of its high boiling point. Therefore, a higher degree of vacuum is required. However, we still confirm that in our sustainable strategy, the IL can be recycled, which can be reused by adding the necessary amount of fresh solvent. 

Figure 7b shows the mechanical properties of our BC composite fibers compared with those of other composite fibers reported in the literatures [49,50,51,52,53,54,55,56,57,58,59,60]. The data of our study are shown in the yellow ellipse. Our fiber can cover a wide range in both tensile strength and Young’s modulus, suggesting that we can control the fiber performances by adjusting the spinning parameters and stretching ratio. In addition, the existing cellulose derivatives in BCNF can act as bridges for combination with other functional polymers or molecules. This type of BCNF can be dispersed in methanol, acetone, and other organic solvents, which is advantageous for composites of BCNF and other polymers. We expect that our strategy can be applied to the fabrication of novel functional fibers based on natural BCNF with special molecules.

## 3. Materials and Methods

### 3.1. Materials

The BCNF slurry was obtained from Kusano Sakko Co. (Hokkaido, Japan) and freeze-dried before use. BCNFs were prepared by rotating or agitating cultures in Hestrin and Schramm’s medium supplemented with some water-soluble cellulose derivatives. BCNFs fermented with hydroxypropyl cellulose (HPC), carboxymethyl cellulose (CMC), and hydroxyethyl cellulose (HEC) were denoted as HPBC, CMBC, and HEBC, respectively. During fermentation, these cellulose derivatives incorporated on the BC microfibril surface via adsorption and prevent the aggregation of newly produced BC. In HPBC, the content of HPC (degree of substitution, DS = 2.8) is 26.0 wt% and the diameter of HPBC is 42 ± 8 nm. In CMBC, the content of CMC (DS = 0.72) is 13.7 wt% and the diameter of CMBC is 27 ± 7 nm. In HEBC, the content of HEC (DS = 2.0) is 22.5 wt% and the diameter of HEBC is 33 ± 7 nm. The lengths of three BCNFs are all in the micron range. More details can be checked in previous article [46].

1-Ethyl-3-methyl-imidazolium acetate (EmimAc) was purchased from Kanto Chemical Co., Inc. (Tokyo, Japan). DMSO and chitosan (CS) (degree of deacetylation ≥ 75%, molecular weight: 190–375 kDa) were purchased from Sigma-Aldrich Co., LLC. (St. Louis, MO, USA). Methanol, ethanol, and acetone were purchased from Kanto Chemical Co., Inc. (Tokyo, Japan). All the chemicals were used without further purification.

### 3.2. Dry Jet-Wet Spinning of HPBC

The HPBC macrofibers were fabricated using dry jet-wet spinning process as follows. HPBC freeze-dried powder was first dissolved in an EmimAc/DMSO cosolvent at 70 °C. The solid content was 4 wt%. The weight ratio of EmimAc and DMSO was 80:20 (*w*/*w*). After the cellulose solution became clear and homogeneous (about 4 h), it was transferred to the cylinder of spinning machine (AIKISSLINE001, AIKI RIOTECH, Aichi, Japan). Then, the dope was extruded from a spinneret with a diameter of 500 μm under 70 °C, passed through an air gap of 6 cm, and coagulated into a methanol bath. The obtained fibers were washed with deionized water for 2 days to remove any residue IL. The water was changed 3 times a day. After washing, the fibers were subjected to wet stretching. The extrusion rate was varied from 15 to 30 m min^−1^. The spinning parameters of the partial samples are listed in Table 1.

### 3.3. Wet Stretching of Macrofibers

After washing, wet stretching of macrofibers was performed using two rolls. The first roll had a constant winding speed (*v*_0_) of 1.5 m min^−1^ and that of the second roll (*v*_1_) ranged between 1.7 and 2.2 m min^−1^. The stretching ratio was calculated using the following formula:$$stretching\;ratio=\frac{v_{1}-v_{0}}{v_{0}}$$

The HPBC fibers obtained with 0, 13, 26, 40, and 46% strains are denoted as HPBC-0, HPBC-13, HPBC-26, HPBC-40, and HPBC-46, respectively. After stretching, the HPBC fibers were dried under tension at 25 °C in air.

### 3.4. Dry Jet-Wet Spinning of CMBC and HEBC

The fabrication process for the CMBC and HEBC macrofibers was the same as described above. However, the washing bath was replaced with a methanol bath.

### 3.5. Dry Jet-Wet Spinning of CMBC and Chitosan (CMCS Fiber)

The preparation process for the CMCS fibers was similar to that described above. The CMBC freeze-dried and CS powders (9:1) were dissolved in an EmimAc/DMSO cosolvent at 95 °C for 14 h. Total solid concentration of the above solution was 4 wt%. After all the biomass was dissolved, the solution was transferred to the spinning cylinder and was spun into methanol at 95 °C. The obtained fibers were washed with water, wet-stretched and dried at 25 °C in air.

### 3.6. Recycling of IL

The coagulation and washing baths were mixed in a flask and transferred to a rotary evaporation equipment. The heating temperature was set to 50 °C. Interior pressure of the flask gradually decreased from 290 to 25 hPa. The process of evaporation continued until no more droplets were observed. The mixture was then transferred to a small flask and subjected to vacuum distillation. The distillation temperature was maintained at 60 °C and the pressure was decreased to 40 Pa. Finally, the residual liquid was collected and prepared for ^1^H NMR test.

### 3.7. Characterization

The viscosity of the HPBC spinning dope was measured using a viscometer (TV-25, Toki Sangyo). The tests for viscosity were performed under different rotation speeds (0–100 rpm) and temperatures (25–80 °C). Attenuated total reflection (ATR)-mode FT-IR were recorded using Nicolet iS10 (Thermo Fisher Scientific Inc., Waltham, MA, USA) in the wavenumbers range from 4000 to 500 cm^−1^. The resolution was 4 cm^−1^ and the number of accumulated scans was set to 64. The crystal structures and indices were determined by XRD analysis using D2 Phaser 2nd Gen diffractometer (Bruker, Karlsruhe, Germany) with Cu-Kα radiation (λ = 0.154 nm). The surfaces and cross-sections morphologies of the HPBC macrofibers were examined through SEM using JSM-6510LV (JEOL, Tokyo, Japan) at an operating voltage of 5 kV. The fiber was sprayed with Pt for 30 s before performing SEM. The alignment morphology was observed through SEM using JSM-7610F (JEOL, Tokyo, Japan). The fiber was cut at an inclined angle in a liquid nitrogen environment and subsequently subjected to a Pt spray treatment before observation. Elemental analyses were performed using a Micro Corder JM10 instrument (J-SCIENCE LAB Co., Ltd., Kyoto, Japan). Tensile tests were performed using an EZ-SX mechanical instrument (Shimadzu Co., Kyoto, Japan), and the mechanical properties were analyzed using the Trapezium X software (1.5.0). The extension rate for tensile test was 5 mm min^−1^ and the gauge length was 58 mm. Unless otherwise specified, the tests were conducted under 21 °C and 40% humidity. Before testing, the samples were placed in a constant temperature and humidity room for 24 h. For the re-wet sample, fibers were immersed in water for 5 min, and then were subjected to mechanical tests immediately under 70% humidity. The cross-sectional areas were measured using optical microscopy (VHX-7100, KEYENCE Co., Osaka, Japan). ^1^H NMR spectra were recorded on a JNM-ECA600 (JEOL, Tokyo, Japan) instrument using D_2_O as the solvent. An approximately 7 mg sample was dissolved in 0.7 mL of D_2_O for the analysis.

## 4. Conclusions

In this study, we used a dry jet-wet spinning method in cooperation with wet stretching to fabricate BC composite macrofibers and investigated the effects of processing parameters on fiber performance. BCNFs can be easily spun into macrofibers using EmimAc as a solvent and the mechanical properties can be enhanced by wet stretching. In this process, the BCNF were assembled into a regular alignment and dense structure through the dissolution by IL and orientation by stretching, as indicated by SEM. This ordered arrangement of BCNF effectively contributed to the transition of the excellent mechanical properties of BC from microscopic to macroscopic level and enhanced the strength and the modulus of the macrofibers. Moreover, IL can be utilized as a “container” to mix BCNFs with other polysaccharides due to the excellent dissolving ability of IL to a variety of polysaccharides. The co-spinning of CMBC with chitosan was confirmed. The intrinsically incorporated cellulose derivatives can increase interactions and compatibility of BCNF with other molecules for further functionalization. IL can be recycled and reused, demonstrating the sustainability of this method. This study provides a scalable strategy for producing functional cellulose composite fibers and promotes the development of a green future with low-carbon emissions.

## Data Availability

The data presented in this study are available in this Manuscript and Appendix A.

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
