# Peer review of "Force-Induced Alignment of Nanofibrillated Bacterial Cellulose for the Enhancement of Cellulose Composite Macrofibers"

_ijms, 2023, doi:10.3390/ijms25010069_

Round 1

Reviewer 1 Report

Comments and Suggestions for Authors

In this manuscript the authors prepared bacterial cellulose-based fibers that contained number of other polysaccharides such as hydoxypropyl cellulose, chitosan etc. The fibers were produced by wet spinning and subjected to various degrees of stretching . The effect of numerous parameters (rotation speed, humidity, spinerette diameter etc) on the mechanical properties of the produced fibers was examined. This manuscript is well-written. The procedures and results are clearly presented and appropriate discussions have been included. The conclusions are supported by the presented results. I suggest acceptance in the present form.

Author Response

Thank you very much for taking the time to review this manuscript and for your kind acknowledgements.

Reviewer 2 Report

Comments and Suggestions for Authors

This paper definitely shows interesting way of spinning bacterial cellulose and improving its strength by alignment using stretching

1.      Line 178 Please add crystallinity% after fiber stretching.

2.      Line 189 Please add diameter and SEM image of fiber after stretching.

3.      Line 198 How long the fibers were soaked in methanol and water bath to remove IL?

4.      Section 3.2 Please add time required to dissolve the cellulose in IL+DMSO

5.      Section 3.2 How much cellulose powder was dissolved ? How was this optimized? Please add solution % or cellulose concentration used.

Author Response

Thank you very much for taking the time to review this manuscript. We have studied your helpful suggestions and carefully revised our manuscript. Please find the detailed responses in the attachment.

Reviewer 3 Report

Comments and Suggestions for Authors

Comments to Takahashi et al

Summary

The study concerns bacterial nanocellulose, as an alternative to petroleum-based polymers employing dry jet-wet spinning and stretching to improve the mechanical properties of produced macrofibers. Furthermore, the authors studied the alignment of the nanofibrils by means of scanning electron microscopy, evaluated the crystalline structure through X-ray defraction and experimentally tested the tensile properties of the fibers. Consequently, the authors evaluated the spinnability and the effect of the process parameters on the tensile properties of the macrofibers. In addition, the study includes the co-spinning of bacterial nanocellulose together with other polymers as well as the recycling of the ionic liquid used as solvent. Based on experimental results, the authors conclude the methods to be scalable to larger production.

General comments

The topic of the study is within the scope of the IJMS journal. Moreover, the manuscript contains the usual sections of a scientific article and the division into subsections improves the readability. In addition, the scientific methods are appropriate and the results distinct with some practical interest. The figures and the tables are of good quality and that applies also to the English language.

Specific comments

Figure 1: It would be helpful to convert the rotation speed to corresponding shear rate values as the conversion depends on the geometry of the viscometer.

Line 128: …rheological properties…

Table 1: Maybe it is not necessary to give constant quantities such as concentration and temperature own columns in the table; it suffices to mention it in the caption or a footnote.

Figure 4 (b): This rather looks like a group of classical stress-strain curves. Hence, should the y-axis be Stress? After all, the (tensile) strength is just a single value for each sample were the fiber breaks and not a continuum as in the graph. The same comment applies for Figure 5 (d).

Lines 351-352: Looking at Figure 7(b), I do not really understand this comment. If your fibers are the ones inside the yellow ellipse, they do not seem to be stronger (higher y-axis values) than most of the other composite fibers. Admittedly they have relatively high Young’s moduli (x-axis), indicating a stiff material.

Lines 387-389: Does the distance between the rolls have any effect? If the deformations are plastic, maybe additional deformation would take place over a longer distance.

Author Response

(The authors gave the same response as above.)

Reviewer 4 Report

Comments and Suggestions for Authors

The works of Wang and coworkers have used dry jet-wet spinning and stretching methods and fabricated cellulose composite macrofibers using nanofibrillated bacterial cellulose (BCNFs) that were obtained by agitated fermentation. The authors have observed that their study provides a scalable method to fabricate bacterial cellulose composite fibers, which can be applied in the textile or biomaterial industries with further functionalization. While the study might be interesting to the audience of the field, the quality of writing is poor. I aimed at seeing a highly improved ms after being processed by a native English writer. My other comments are summarized below.

1.     The introduction of the paper is a bit tiring, thus squeezing the paper to a maximum of one page with appropriate focus and motivation is imperative.

2.     The FTIR spectra need in-depth description. I am not sure whether the assigned peaks (marked by lines in Fig. 6a) represent noises or real absorption. What was the resolution of the spectra?

3.     Young’s modulus is a good physical descriptor. The results are well described. However, the observed data are not verified by simulation. To this end, I am not sure whether the data reported are genuine and can be reproducible!!

Comments on the Quality of English Language

--

Author Response

(The authors gave the same response as above.)

Round 2

Reviewer 4 Report

Comments and Suggestions for Authors

Authors of this interesting work have considered my comments and have revised their paper. 
I believe that this Ms is ready for acceptance.